# The COVID-19 Era—Influencers of Uneven Sector Performance: A Canadian Perspective

**Vikkram Singh** [1],*, **Homayoun Shirazi** [2] **and Jessica Turetken** [3]

1   Department of Global Management, Ted Rogers School of Management, Ryerson University, Toronto, ON M58 2K3, Canada
2   Department of Economics, Ryerson University, Toronto, ON M58 2K3, Canada; homayoun.shirazi@ryerson.ca
3   Ted Rogers School of Management, Ryerson University, Toronto, ON M58 2K3, Canada; jessica.turetken@ryerson.ca
*   Correspondence: vik.singh@ryerson.ca

**Abstract:** The study estimates the impact of COVID-19 on the labour market outcomes of major industrial sectors in Toronto, the largest urban centre in Canada. Using various economic data, we classify the sectors as distressed, stable, and those requiring ongoing monitoring. Furthermore, we estimate the expected impact of the pandemic shock using the Impulse Response Function (IRF) method. The results show an uneven impact of the pandemic with adverse outcomes for low-paying front-facing sectors, such as accommodation & food services and manufacturing. The post-pandemic projections show lingering negative implications for various sectors. The insights are helpful for policy recommendations, such as targeted responses to address the declines and structural changes in these sectors because of increasing technology adoption and the resulting labour market challenges.

**Keywords:** COVID-19; Canada; labour market; industrial sectors

## 1. Introduction

The COVID-19 pandemic has profoundly affected the global economy along with causing grave implications for public health (Gormsen and Koijen 2020; Ingravallo 2020; Ozili and Arun 2020). The socio-economic impact has been uneven, as vulnerable segments of the population, such as racialised individuals, women, and those with less education, face the brunt of the pandemic (Milani 2021; Béland et al. 2020; Blundell et al. 2020; Gupta et al. 2020; Rojas et al. 2020; UN 2020; Yasenov 2020). Several key themes have emerged that guide our reflection and understanding of the impact of COVID-19 and the ability to predict future scenarios. Apart from the economic disruption and welfare losses resulting from adverse health outcomes, the pandemic has profoundly changed consumer behaviour. In the initial periods of the pandemic, consumers followed health concerns by changing the way they shopped, moving aggressively towards online shopping and limiting purchases that were considered essential (Andersen et al. 2020; Ceylan et al. 2020; Goolsbee and Syverson 2020). Several sectors of the economy, such as retail, hospitality, and travel & tourism, continue to experience significant disruptions, with the resulting organisational changes expected to last well beyond the pandemic. The effects of the pandemic are likely to shift the economic trajectory, with the recovery from the downturn being uneven. Many sectors will continue to struggle, while others will likely grow beyond pre-pandemic levels.

This study seeks to identify emerging trends that are likely to continue in the post-pandemic phase and transform the labour market and industrial sectors in Toronto, the largest urban centre in Canada. The research question we ask is: How has COVID-19 influenced the labour market of major sectors of the economy in Toronto? The motivation for this study stems from the unequal impact of the pandemic on the economy and its sectors. Addressing this question and identifying the potential causes of the differential effects of the pandemic on different sectors can help make predictions and, consequently,

policy suggestions for the post-pandemic period. We first estimate the impact of COVID-19 on the economy and the resulting labour market trends. Then, we analyse the top industry sectors, assessing the economic fallout due to the pandemic and provide forecasts of how they are likely to evolve in the post-pandemic economy. Our study offers several contributions. The findings can contribute to policymaking; the initial identification of industry and organisational trends can spark strategic conversations about potential interventions needed to change the current revenue tools and policy frameworks in the post-pandemic era. The study is structured as follows. Section 2 provides a brief literature review, while Section 3 explains the methodology. Section 4 presents the results, and Section 5 concludes with a discussion and conclusion.

## 2. Literature Review

The industrial transformations resulting from the pandemic come with an extensive list of inefficiencies and structural problems. Some sectors of the economy are likely to go through a labourious process of reassessing their core structures. Traditional industries will need to seek new operational frameworks to provide short-run confidence, including remote-work flexibility and the re-evaluation of key supply chains. Notably, while the pandemic has touched every industry, the challenges are not uniform. Studies in other jurisdictions show a profound impact on employment in specific sectors, such as accommodation and food services, retail, real estate, and other personal services (Svabova et al. 2020; Almeida and Santos 2020; Byrne et al. 2020; Bartik et al. 2020; Harris et al. 2020; Rapaccini et al. 2020). In Canada, the initial impact of the pandemic was severe, with a 32% decline in aggregate weekly hours for workers in the age category of 20–64, leading to an overall 15% decline in employment (Lemieux et al. 2020).

Deng et al. (2020) find that only 40% of employed Canadians have even a slight possibility of working remotely. However, despite the lack of agility to transform the workforce, the study finds observable changes in some industries, such as financial services, prevented a worst-case scenario for the Canadian labour market. Although there are considerable operational differences in the flexibility and ability of sectors to offer remote work, there are often overlooked aspects of industries, such as their ability to adapt quickly to the rethinking of the workplace during the pandemic. As Adams-Prassl et al. (2020) note, there is significant variability and heterogeneity within industries regarding employee ability to work remotely—the effect is comparatively more important than the observed differences in adaptability between sectors. For example, retail giants such as Walmart, Amazon, and Costco were financially and operationally insulated compared to their smaller competitors. While labour supply shifts beneath them, they can capitalise on the rest of the demand left in the wake of their smaller competitors, potentially going out of business because of the pandemic. Barrero et al. (2020) talk about this phenomenon and note that from March to May 2020, mega-businesses hired between three and four new employees for every ten layoffs in North America due to COVID-19. However, the pandemic has had a disproportionate impact on labour market outcomes on specific segments of the population, such as females (UN 2020), youth (Churchill 2021; Svabova and Kramarova 2021) and older individuals (Svabova and Gabrikova 2021) leading to societal inequalities.

Another significant theme evident during the pandemic is the increasing technology adoption in industries, also known as Industry 4.0.[1] According to Rymarczyk (2020), IR 4.0 refers to the prominence of a range of emerging technologies, including but not limited to big data, cloud computing, artificial intelligence, the internet of things, advanced machines, and blockchain. New technologies provided resiliency in supply chains and cost savings that were pivotal in the survival of many economic sectors.[2] While the mass replacement of labour due to automation sounds grim in an already damaged labour market, this situation does not have to be the ultimate fate of industries in a post-pandemic world. Shestakofsky (2017), who analysed 19 months of industry and technology co-evolution, observes a sharp shift to labour involving more computational and technological skills rather than full automation of work. Lichtenthaler and Fischbach (2019) recalls that the basis of technology

implementation is to move towards the ultimate goal of competitive advantage—while some artificial intelligence technologies can offer immediate operational benefits, there is a need to create a technologically positive infrastructure. While these studies put forward the changes in the industrial landscape, none provide a deeper understanding of how the pandemic changed the sectors, especially their labour market outcomes in Toronto. Our study fills this key literature gap.

## 3. Methodology

Our study seeks to identify and classify sectors based on their stability level to understand the impact of COVID-19 across sectors within Toronto. To that end, we use the following survey data: Canadian Labour Force Survey (LFS), Canadian Survey of Business Conditions (CSBC) and Longitudinal Employment Analysis Program (LEAP) from Statistics Canada. To forecast[3] the impact of COVID-19 on various sectors, we employ a $4 \times 4$ VAR model[4], which uses quarterly data from 2000q1 to 2020q3. Such a model permits the dynamics of shocks—impulse response functions (IRFs)—imposed by different variables on unemployment by sector. Due to data availability, the IRFs are estimated using provincial data. The VAR model includes seasonal dummies as exogenous variables to address seasonality. Cholesky ordering is used to estimate the impulse response functions (IRFs):[5]

$$Un_t = a_0 + \sum_{i=1}^{k} a_{1i}Un_{t-i} + \sum_{i=1}^{k} b_{1i}GDPSector_{t-i} + \sum_{i=1}^{k} c_{1i}GDPG_{t-i} + \sum_{i=1}^{k} d_{1i}Inf_{t-i} + \sum_{i=1}^{3} f_{it}Dum_{it} + e_{1t} \qquad (1)$$

$$GDPSector_t = b_0 + \sum_{i=1}^{k} a_{2i}Un_{t-i} + \sum_{i=1}^{k} b_{2i}GDPSector_{t-i} + \sum_{i=1}^{k} c_{2i}GDPG_{t-i} + \sum_{i=1}^{k} d_{2i}Inf_{t-i} + \sum_{i=1}^{3} f_{it}Dum_{it} + e_{2t} \qquad (2)$$

$$Inf_t = d_0 + \sum_{i=1}^{k} a_{4i}Un_{t-i} + \sum_{i=1}^{k} b_{4i}GDPSector_{t-i} + \sum_{i=1}^{k} c_{4i}GDPG_{t-i} + \sum_{i=1}^{k} d_{4i}Inf_{t-i} + \sum_{i=1}^{3} f_{it}Dum_{it} + e_{4t}) \qquad (3)$$

$$GDPG_t = c_0 + \sum_{i=1}^{k} a_{3i}Un_{t-i} + \sum_{i=1}^{k} b_{3i}GDPSector_{t-i} + \sum_{i=1}^{k} c_{3i}GDPG_{t-i} + \sum_{i=1}^{k} d_{3i}Inf_{t-i} + \sum_{i=1}^{3} f_{it}Dum_{it} + e_{3t}) \qquad (4)$$

where, $Un_t$ denotes unemployment rate in each sector in Ontario; $GDPSector_t$ represents the GDP growth rate of each sector; $Inf_t$ indicates the aggregate inflation rate for Ontario; $GDPG_t$ represents the growth rate of Ontario's economy, $Dum_i$ denotes an exogenous seasonal dummy variable, and $e_t$ is a random error with 0 mean and constant variance. The response from the shock is measured starting the first quarter of 2021, and the total period for the response is ten quarters. The GDP data are from the Ontario Ministry of Finance, and the unemployment rate is derived from Statistics Canada for the first quarter of 2000 to the third quarter of 2020. Please note that the employment data frequency is monthly, while the IRFs use quarterly periods.

We classify sectors according to the effects of COVID-19.[6] The top industries or sectors are identified according to the significance of their share of total employment in Toronto (Table 1) and classified according to the criteria listed in Table 2 [7]. To identify sectors in the "distressed" category, we use two measures—an adverse change in year-over-year (2020–2019) employment along with eight or more month-to-month declines in employment (Table 2). Such sectors not only faced immense adverse effects during the pandemic, but are also likely to take the longest to recover in the post-pandemic period. Their recovery will likely require targeted assistance and a policy framework for structural reforms. Sectors classified as "must be monitored" are those with an adverse change in year-over-year employment (2020–2019) with a 3 to 8 month decline in employment. While not in crisis, these sectors will need ongoing monitoring and limited help to recover from the pandemic. Finally, those in the "stable" category experienced a positive change in employment over 2019–2020 and did not require targeted assistance.

**Table 1.** Top 10 sectors by employment (Toronto).

|  | 2016 | 2017 | 2018 | 2019 | 2020 |
|---|---|---|---|---|---|
| Professional, scientific and technical services | 12% | 13% | 13% | 13% | 14% |
| Finance, insurance, real estate & leasing | 12% | 12% | 11% | 11% | 12% |
| Healthcare & social assistance | 11% | 11% | 11% | 12% | 12% |
| Retail trade | 9% | 10% | 10% | 9% | 10% |
| Educational services | 7% | 6% | 7% | 8% | 8% |
| Manufacturing | 9% | 8% | 8% | 8% | 7% |
| Construction | 6% | 6% | 6% | 6% | 6% |
| Accommodation & food services | 7% | 8% | 7% | 6% | 5% |
| Transportation & warehousing | 3% | 4% | 5% | 5% | 5% |

Source: Labour Force Survey, Statistics Canada.

**Table 2.** Sector Classification criteria.

| Category | Criteria |
|---|---|
| Distressed | Negative change in the year over year employment levels (2020–2019) and 8 or more negative monthly year over year (2020–2019) decline in employment |
| Must be monitored | Negative change in the year over year employment levels (2020–2019) and 3–8 negative monthly year over year (2020–2019) change in employment |
| Stable | Positive change in the year over year employment levels (2020–2019) |

## 4. Results

Table 3 illustrates the placement of the major sectors of Toronto's economy in various categories based on the criteria listed in Table 1. Amongst all the sectors studied, the accommodation & food services and manufacturing sectors fared the worst, with losses of 31% and 10% in employment in 2020, respectively. Furthermore, they exhibited persistent negative sentiments—eight or more monthly employment declines—classified as "distressed." The retail, transportation & warehousing, health care & social assistance, construction, and educational services sectors are placed in the "must-be monitored" category—despite an annual decline in employment in 2020, they show a revival in employment numbers towards the later part of the year. Last, the financial, insurance, real estate & leasing and professional, scientific & technical services sectors are classified as "stable" because of their robust employment structure and a positive year-over-year change in employment.

**Table 3.** Sector Classification.

| | 2016–2017 | 2017–2018 | 2018–2019 | 2019–2020 | # of Months of Negative Change | Category | Negative IRF Periods (Quarters) | Is the Impulse Persistent? |
|---|---|---|---|---|---|---|---|---|
| | (1) | (2) | (3) | (4) | (5) | (6) | (7) | (8) |
| Accommodation & food services | 14% | −8% | −9% | −31% | 12 | Distressed | 5 | No |
| Manufacturing | −8% | 9% | −8% | −10% | 8 | Distressed | 9 | No |
| Retail trade | 4% | 7% | −8% | −6% | 5 | Must be monitored | 8 | No |
| Transportation & warehousing | 20% | 14% | 16% | −19% | 4 | Must be monitored | 3 | No |
| Healthcare & social assistance | −1% | −0.4% | 11% | −8% | 4 | Must be monitored | N/A | No |
| Construction | −1% | −3% | 8% | −8% | 3 | Must be monitored | 4 | No |
| Educational services | −8% | 10% | 13% | −9% | 3 | Must be monitored | 6 | No |
| Finance, insurance, real estate & leasing | −4% | −7% | 7% | 2% | 1 | Stable | 9 | No |
| Professional, scientific & technical services | 5% | 4% | 2% | 1% | 1 | Stable | 5 | No |

Notes: (1) Columns 1 to 4 depict the annual year-over-year change in employment. (2) Column 5 depicts the number of months of negative employment change in 2020. (3) Column 6 depicts the category where the sector falls using the criteria listed in Table 1. (4) Column 7 depicts the number of negative quarters of response to the COVID-19 shock starting after the first quarter of 2021, using the Impulse Response Functions (IRF). Due to lack of data, IRF is not estimated for the healthcare & social assistance sector. Provincial data is used to estimate the IRFs. (5) Column 8 indicates if the response from the COVID-19 was persistent (remained negative) over the ten quarters starting from the first quarter of 2021.

*4.1. Distressed Sectors*

4.1.1. Accommodation & Food Services

This sector comprises establishments primarily engaged in providing short-term lodging and the preparation and sale of food.[8] The COVID-19 restrictions around dining and the restrictions and fears around travel heavily impacted this sector, with most of the workforce unable to pivot to remote work—only 7% of the workforce transformed to remote work due to the pandemic (Table 3). The demographics of employees tend to be less educated and likely to be youth because of the flexibility in scheduling—approximately 62% were between the ages of 15–24 and working part-time.[9] Likewise, the average hourly wage in this sector was in the lower quartile compared to other industries—the provincial average salary was $17.07 in 2020, the lowest compared to other sectors.[10] More businesses closing than opening also depict sectoral weakness—in 2020, there was a net loss of 1327 businesses in this sector in Toronto (Figure A2, Appendix A). Business closing as a percentage of active businesses was 8%, almost double the rates in the previous two years (Table A1, Appendix A).

Employment in this sector declined by 31% in 2020 compared to the previous year—the largest drop among all major sectors (Table 2). This sector was especially vulnerable because of the weakness in job numbers before the pandemic. Employment trended down, with declines of 8% and 9% in 2017–2018 and 2018–2019, respectively. Trends in 2020 show that employment in all months was lower than in similar months in the previous three years due to the onset of COVID-19 and the early lockdown (Figure 1). Employment fell by 40% in March, the highest among all major sectors, and by another 17% in April. With the reopening in July, the number rose again, with a peak in September of 73,000, but fell again in October and December because of the lockdown in response to the pandemic's second wave. Overall, this sector lost 349,000 jobs in 2020, ranking the most impacted by

the pandemic, led by losses in its largest subsector—food services and drinking places (Figure 2). A shift in retail spending on food expenditure from restaurants to home cooking was likely a contributor to the pain experienced by this sector (Goddard 2020). The IRF forecast starting the first quarter of 2021 shows that a negative shock is likely to increase unemployment for five quarters (Figure 3 and Table A2, Appendix A). However, the future is expected to be volatile, with a recovery from a decline in employment expected only in the ninth quarter. Other countries such as Slovakia (Svabova et al. 2020), Portugal (Almeida and Santos 2020), Ireland (Byrne et al. 2020) and the US (Bartik et al. 2020) also reported a rapid decline in employment in this sector due to the pandemic.

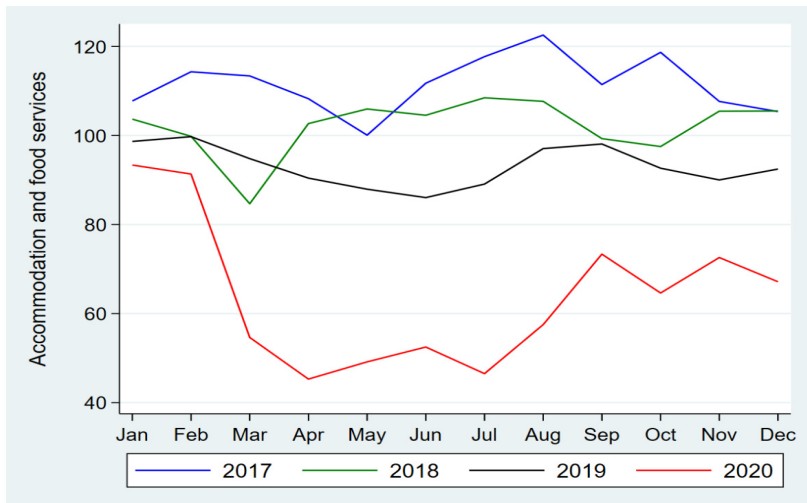

**Figure 1.** Employment in the Accommodation and Food Services sector. The employment in all months in 2020 was lower than the corresponding months in previous years. The y-axis of the figures represents the number of employees in 1000s. Source: Labour Force Survey, Statistics Canada.

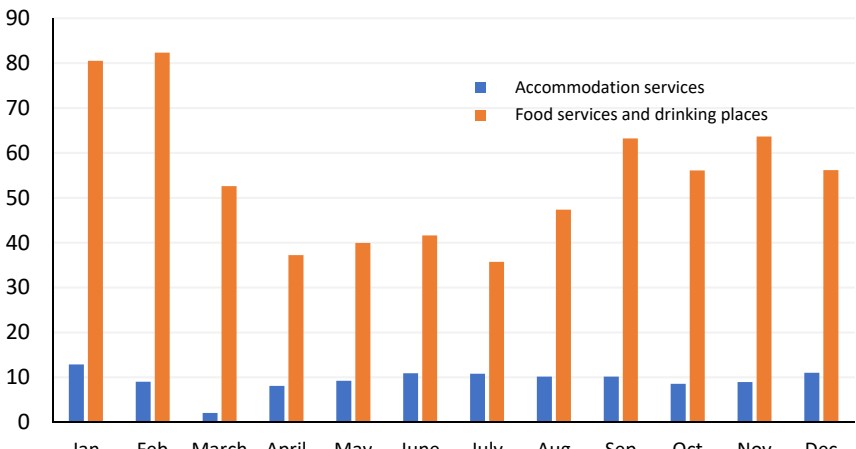

**Figure 2.** Accommodation and Food Services—employment in key segments, (2020). Employment in food services & drinking places struggled to recover in the post-pandemic period compared to accommodation services. The y-axis of the figures represents the number of employees in 1000s. Source: Labour Force Survey, Statistics Canada.

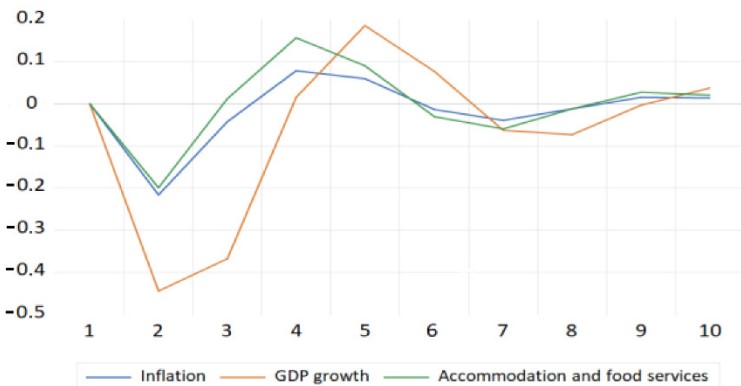

**Figure 3.** Accommodation and Food Services—IRF[11]. The figure depicts the impact of inflation, GDP growth for Ontario, and GDP in the accommodation and food services sector on unemployment in the sector, starting the first quarter of 2021. The effect (response) of the shock (COVID-19) was negative in the fourth and fifth quarters and did not entirely dissipate until the ninth quarter.

### 4.1.2. Manufacturing

The manufacturing sector comprises establishments primarily engaged in the chemical, mechanical or physical transformation of materials or substances into new products.[12] The trends in this sector were closely related to the shutdowns in the spring and early summer of 2020—four-fifths of manufacturers in Toronto reported a negative impact of COVID-19 on their operations.[13] At the onset of the pandemic restrictions, the yearly month-over-month drop in manufacturing sales was 40% (Table 4). While a rebound took place in the subsequent months, the year ended with a decline in the remaining year. The pandemic took a toll on enterprise viability in this sector, with 3472 businesses opening and 4036 closings, with the bulk of the shutdowns occurring from March to July (Figure A2, Appendix A).

**Table 4.** Manufacturing sales.

| | January | February | March | April | May | June | July | August | September | October | November | December |
|---|---|---|---|---|---|---|---|---|---|---|---|---|
| Month to month change | −2% | 4% | 0% | −40% | 19% | 41% | 1% | 0% | 8% | −1% | 0% | −9% |
| Year over year change | −7% | 1% | −11% | −43% | −39% | −9% | 1% | −7% | 2% | −5% | −1% | 1% |

Source: Monthly Survey of Manufacturing, Statistics Canada.

This sector accounted for 7% of total employment in 2020, down from 8% in the previous year (Table 3). The weakness in manufacturing sales also translated into a depressed job market in 2020, with a loss of 8% compared to the previous year (Table 2). Employment fell to a low of 86,100 in July, with a slight rebound in the subsequent months due to the economy's reopening (Figure 4). A drill-down of the key subsectors indicates that employment in durable goods manufacturing fared better because of the surge in online shopping (Figure 5). Despite this, the monthly employment numbers remained lower than those in the previous three years (Figure 4). The inability of most of the jobs in this sector to transition to remote work hampered the employment market—only 10% of the workforce could transition to remote work (Figure A1, Appendix A). The IRF forecast shows that the negative shock is likely to increase unemployment for at least nine quarters (Figure 6 and Table A2, Appendix A). Other countries, such as the UK (Harris et al. 2020) and Italy (Rapaccini et al. 2020), also face rapid declines in their manufacturing sector.

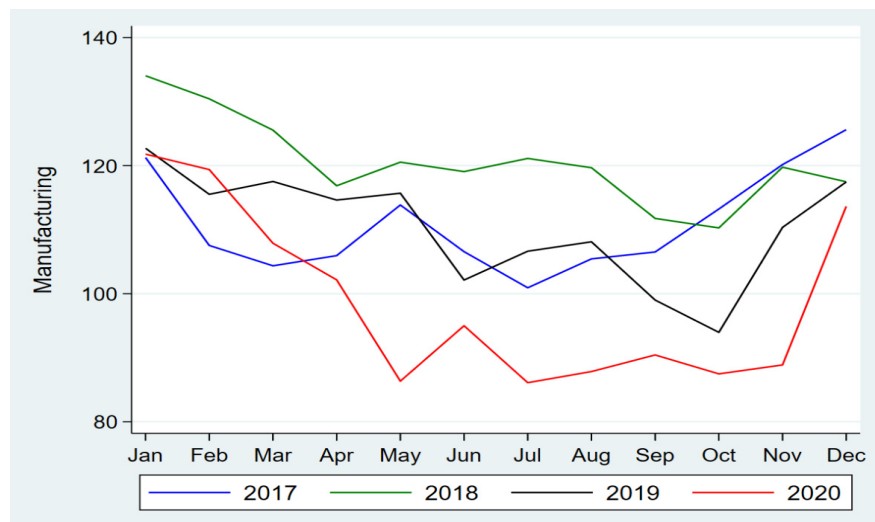

**Figure 4.** Employment in Manufacturing sector. The employment in all months post-March 2020, the start of the pandemic, was lower than the corresponding months in previous years. The y-axis of the figures represents the number of employees in 1000s. Source: Labour Force Survey, Statistics Canada.

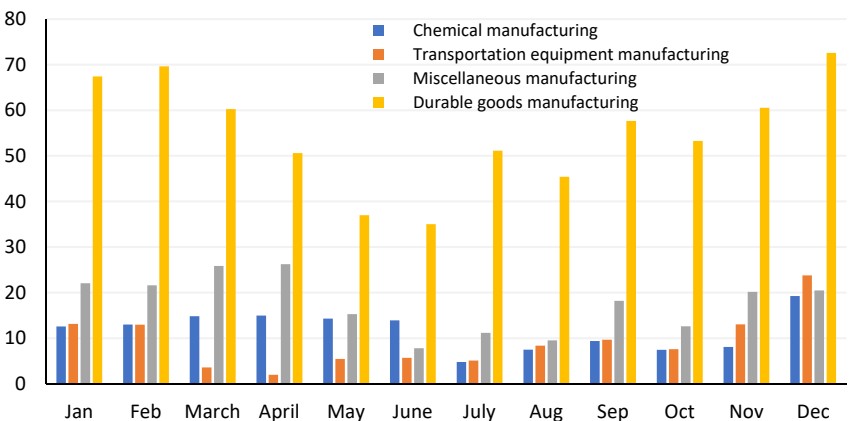

**Figure 5.** Manufacturing—employment in key segments, 2020. The y-axis of the figures represents the number of employees in 1000s. Source: Labour Force Survey, Statistics Canada.

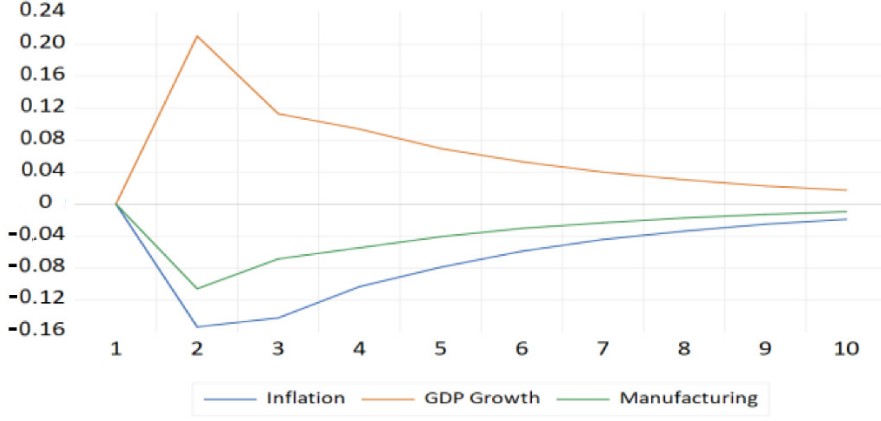

**Figure 6.** Manufacturing—IRF[14]. The figure depicts the impact of inflation, GDP growth for Ontario, and GDP in the manufacturing sector on unemployment in the sector, starting the first quarter of 2021. The effect (response) of the shock (COVID-19) beginning in 2021 is positive for the first three quarters, but shows increasing weakness and converges over time.

*4.2. Must Be Monitored Sectors*

4.2.1. Retail Trade

The retail sector comprises of firms engaged in retailing merchandise, generally without transformation, and rendering services related to merchandise sales.[15] It was the fourth largest sector in Toronto in terms of employment, accounting for 10% of the total workforce (Table 3). The pandemic shutdown immediately impacted retail sales, resulting in a plunge of 40% in April 2020 (Table 5). The front-line facing nature of much of the employment in this sector meant that only 12% of the workforce could transition to remote work (Figure A1, Appendix A). While sales rebounded strongly in the following months, a second COVID-19 wave led to a further decline during the important holiday season in December. The number of businesses that closed as a percentage of active businesses was 7% in 2020 compared to the average of 4% in the previous two years, highlighting the weakness in this sector (Table A1, Appendix A). More businesses closed (9728) than opened (8620) owing to many closings during April, May and June (Figure A2, Appendix A).

**Table 5.** Retail sales in 2020.

|  | January | February | March | April | May | June | July | August | September | October | November | December |
|---|---|---|---|---|---|---|---|---|---|---|---|---|
| Month to month change | −2% | 4% | 0% | −40% | 19% | 41% | 1% | 0% | 8% | −1% | 0% | −9% |
| Year over year change | −7% | 1% | −11% | −43% | −39% | −9% | 1% | −7% | 2% | −5% | −1% | 1% |

Source: Survey of Retail Trade, Statistics, Canada.

The employment market also mimicked the sales drop in April; however, the numbers rebounded in the subsequent months, ending 2020 with an annual decline of only 6% from the previous year (Figure 7 and Table 2). Interestingly, the employment trend shows that December 2020's monthly employment figure was higher than the previous three years. The robust rebound in employment was because of the increase in online retail sales.[16] Monthly employment at the end of 2020 outpaced that of 2017–2019 due to a rise in consumer purchases in response to travel and leisure restrictions.[17] An analysis of major retail subsectors finds a robust increase in employment in food & beverage as well as store retailer segments of this sector (Figure 8). The impulse response to the COVID-19 shock illustrates a persistent negative trend for the next eight quarters, highlighting the vulnerability to lockdowns and changing consumer shopping behaviour (Figure 9 and Table A2, Appendix A).

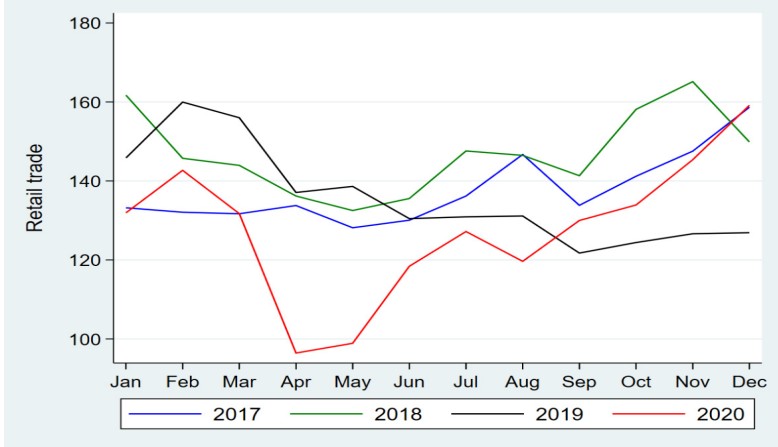

**Figure 7.** Employment in the Retail Trade sector. The employment in months after the pandemic in 2020 declined rapidly, but bounced back starting August. The y-axis of the figures represents the number of employees in 1000s. Source: Labour Force Survey, Statistics Canada.

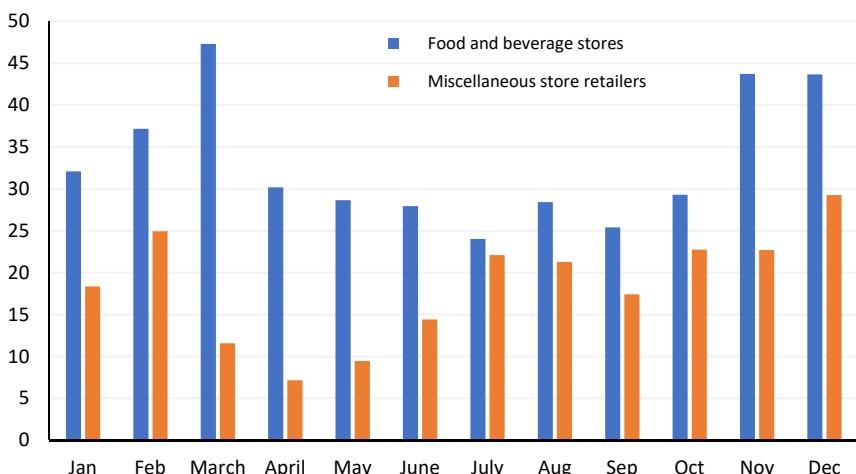

**Figure 8.** Retail Trade sector—employment in key segments, 2020. Unlike miscellaneous store retailers, employment in the food and beverage stores segment remained lower in the period after the start of the pandemic. The y-axis of the figures represents the number of employees in 1000s. Source: Labour Force Survey, Statistics Canada.

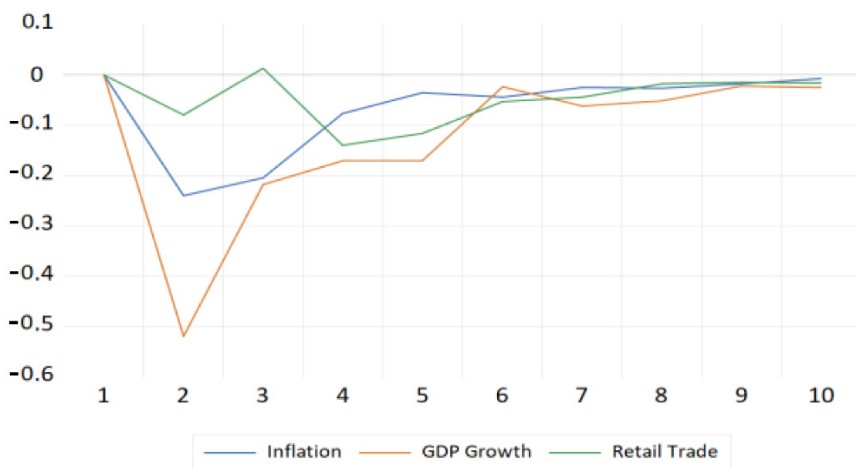

**Figure 9.** Retail Trade—IRF[18]. The figure depicts the impact of inflation, GDP growth for Ontario, and GDP in the retail trade sector on unemployment in the sector, starting the first quarter of 2021. The IRF shows a persistent negative trend that dissipates in the tenth quarter with the convergence.

### 4.2.2. Transportation and Warehousing

This sector includes organisations primarily engaged in transporting passengers and goods, warehousing and storing goods.[19] It employed 5% of the workforce in Toronto in 2020 (Table 3), including many essential workers.[20] While employment in the warehousing segment increased because of the dependence on online shopping and people working from home, the transport segment suffered considerable losses owing to the pandemic.[21] Transit usage in Toronto dropped by 21% since the pandemic's onset due to work-from-home orders and fears associated with public transport (Habib et al. 2021). The rising virus cases in the TTC further depressed transit ridership.[22] Federal and provincial travel bans also affected the industry, with air travel and other tourism-related travels falling significantly during the pandemic.[23] Likewise, business travel decreased owing to many workplace restrictions, leading to a decline in travel revenues (Lundy 2021). The percentage of businesses closing out of the total active businesses was 9% in 2020, the highest among all the major sectors and exceeding the rates of 8% in the previous two years (Table A1,

Appendix A)—net business closures amounted to 977, the third most impacted sector in Toronto.

The employment trends during 2020 indicate a moderate decline in the initial months of the pandemic. Nevertheless, the situation in the subsequent months changed with a significant loss of 10,000 jobs in July (Figure 10). Monthly employment trends also remained lower than in previous years, starting in August. The transit & ground transportation sub-sector experienced a sharp decline in employment mitigated by the revival in employment in truck transportation (Figure 11). The year ended with a 19% drop in employment, the second-largest decline amongst all sectors (Table 2). The IRF forecast for the sector's GDP shows that the initial negative impact of COVID-19 occurs in the second quarter of 2021 (Figure 12 and Table A2, Appendix A). Further declines occur in the sixth and seventh quarters, tapering off in the ninth quarter.

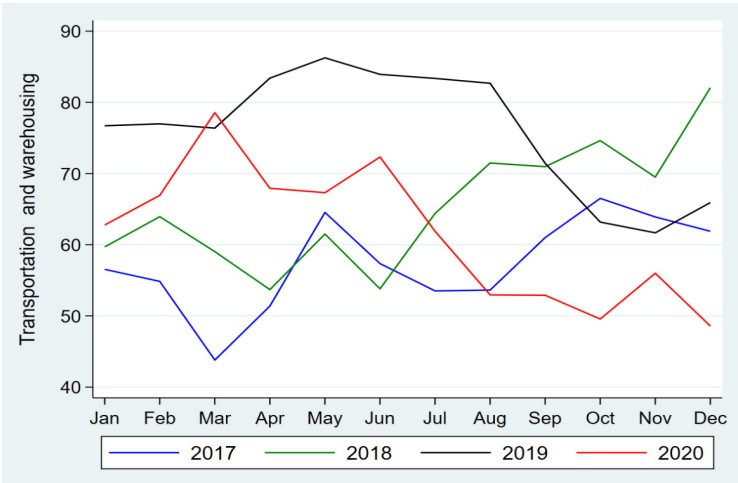

**Figure 10.** Employment in Transportation and Warehousing sector. While employment after the start of the pandemic in 2020 did not decline immediately, a rapid decline occurred from June, with the monthly levels starting in August being lower than those in the corresponding months in previous years. The y-axis of the figures represents the number of employees in 1000s. Source: Labour Force Survey, Statistics Canada.

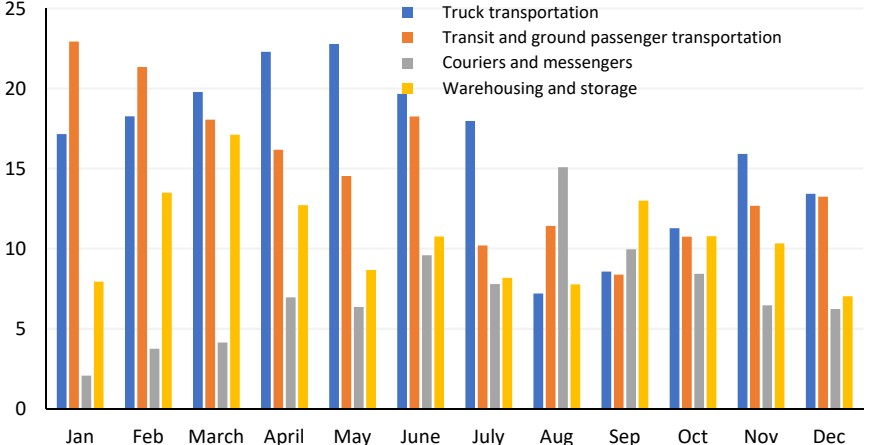

**Figure 11.** Transportation and Warehousing sector—employment in key segments, 2020. The y-axis of the figures represents the number of employees in 1000s. Source: Labour Force Survey, Statistics Canada.

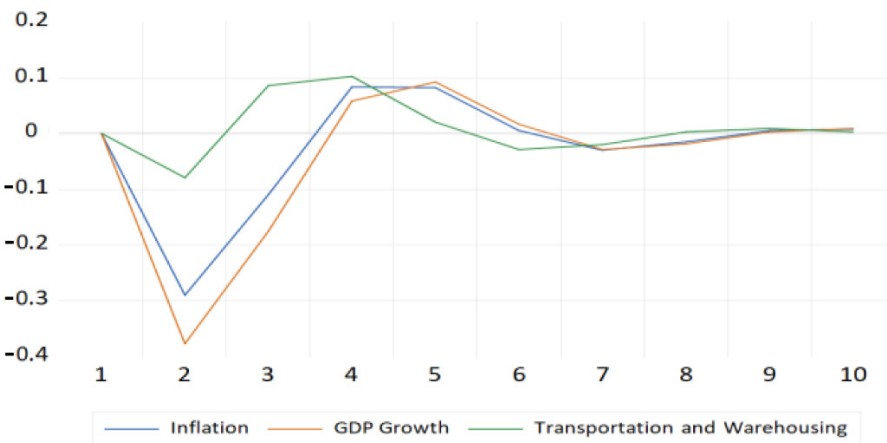

**Figure 12.** Transportation and Warehousing sector—IRF[24]. The figure depicts the impact of inflation, GDP growth for Ontario, and GDP in the transportation and warehousing sector on the sector unemployment, starting the first quarter of 2021. The negative shock lasts for two quarters, further declining in the sixth and seventh quarters and dissipates at the end of the ninth quarter.

### 4.2.3. Healthcare and Social Assistance

This sector consists of establishments primarily providing health care by diagnosis, treatment and social assistance, such as counselling, child protection, community housing and food services.[25] It is the third-largest sector in Toronto, accounting for 12% of total employment (Table 3); it reported an 8% decline in job numbers in 2020 (Table 2). Even though business closures exceeded business openings, the net closures (178) were lower than those in other sectors (Figure A2, Appendix A). The sector was shielded from massive layoffs, as experienced in other sectors, as many workers were classified as essential and benefited from being in the higher quantile of employment wages (Tal 2021).

The drop in employment mainly occurred from April to July, resulting from the COVID-19 lockdowns and the layoffs of registered nurses and administrative staff stemming from budgetary cuts of the province (Figure 13). However, there was an uptick in employment, especially in ambulatory health care and hospitals, because the provincial government added resources to address the critical health care needs related to virus transmission (Figure 14). After July, the revival in employment provided a respite, although the numbers failed to match the pre-pandemic levels.

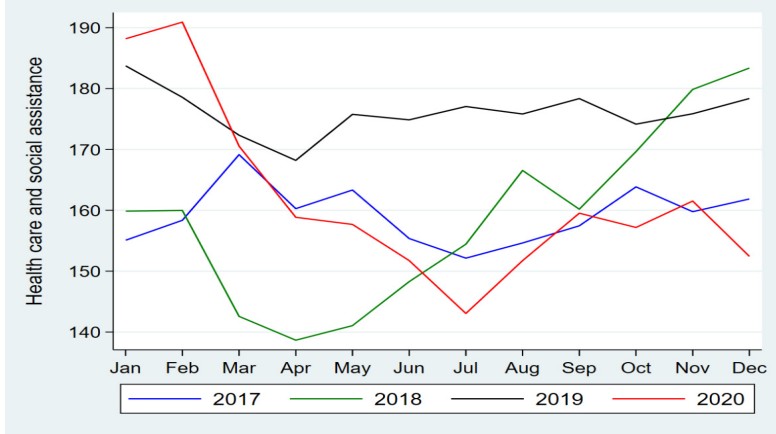

**Figure 13.** Employment in Healthcare and Social Services sector. The employment dropped at the start of COVID-19; however, a revival is noted starting July due to increased levels of government funding to mitigate the impact of the pandemic. The y-axis of the figures represents the number of employees in 1000s. Source: Labour Force Survey, Statistics Canada.

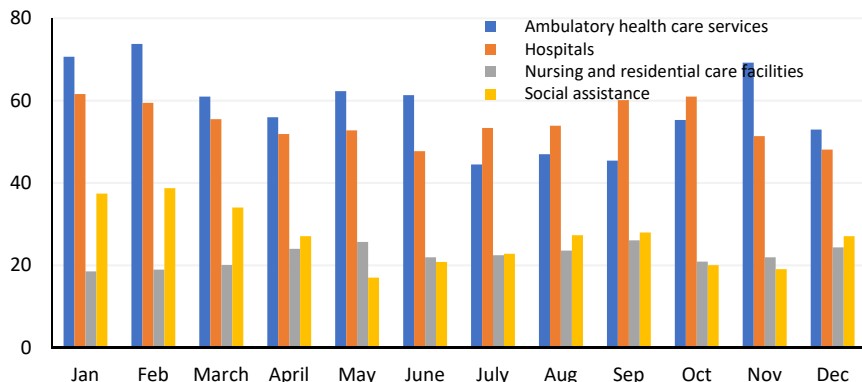

**Figure 14.** Healthcare and Social Services sector—employment in key segments, 2020. The y-axis of the figures represents the number of employees in 1000s. Source: Labour Force Survey, Statistics Canada.

4.2.4. Educational Services

Educational services, including colleges, universities, and training centres engaged in providing instruction and training in various subjects[26], and employed 8% of the total workforce in Toronto in 2020 (Table 3). This sector was relatively unscathed by business closings, with a net closing of 149 in 2020, the lowest among the city's top sectors (Figure A2, Appendix A). However, it was impacted by the pandemic shutdown and transition to online learning as employment fell by 9% in 2020 (Table 2), with sharp declines during February, March, April and May (Figure 15). Even before the pandemic, employment in this sector had been reeling from provincial cutbacks and labour disputes in elementary, secondary, and postsecondary institutions.[27]

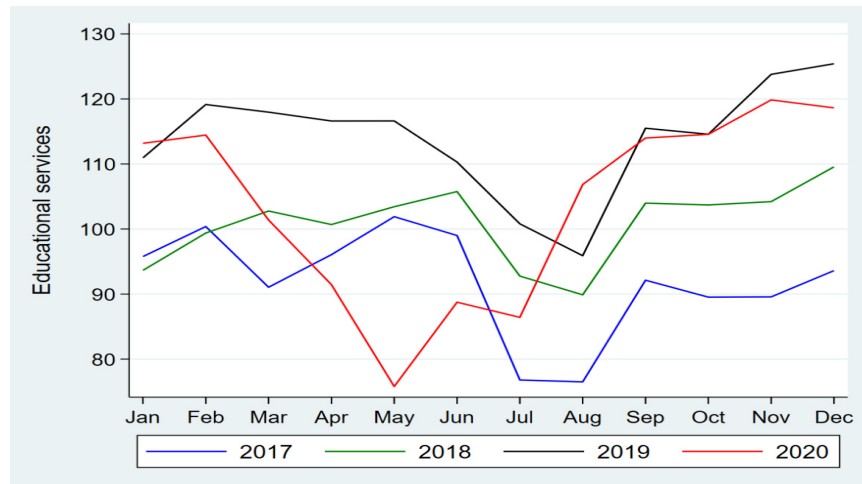

**Figure 15.** Employment in Educational Services sector. While the start of the pandemic led to a rapid decline in employment in April and May, the levels recovered due to the reopening of schools and added funding to meet online learning requirements. The y-axis of the figures represents the number of employees in 1000s. Source: Labour Force Survey, Statistics Canada.

The growth in employment in August 2020 was likely related to the provincial government's efforts to increase headcounts to facilitate smaller class sizes and online education to aid with COVID-19 safety regulations.[28] Most of the additional jobs were in elementary & secondary schools, while community colleges & universities sub-sector ended the year at a net loss in employment (Figure 16). However, the impulse response from the shocks from the pandemic shows six quarters of expected adverse reactions (Figure 17 and Table A2,

Appendix A). Despite a respite in the fifth quarter, further declines occurred in the next three quarters.

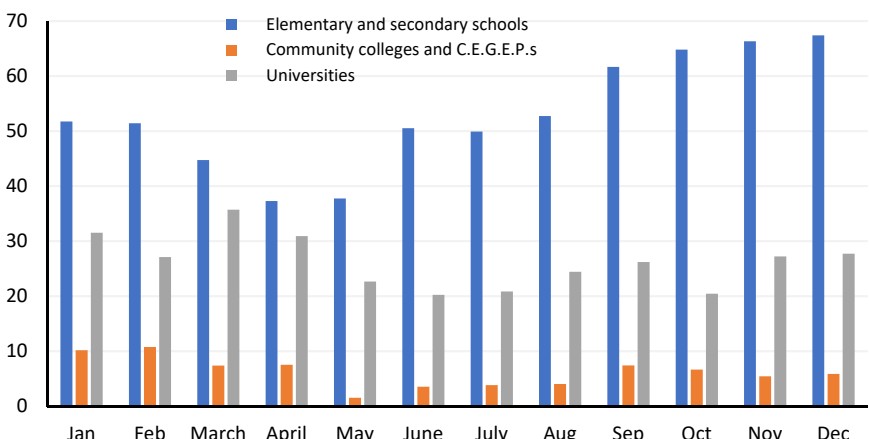

**Figure 16.** Educational Services—employment in key segments, 2020. A strong increase in employment is reported for elementary & secondary schools. The y-axis of the figures represents the number of employees in 1000s. Source: Labour Force Survey, Statistics Canada.

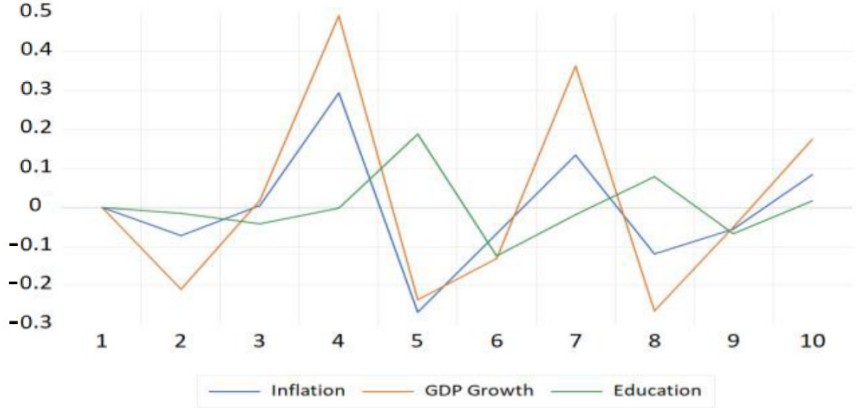

**Figure 17.** Educational Services—IRF[29]. The figure depicts the impact of inflation, GDP growth for Ontario, and GDP in the education sector on unemployment in the sector, starting the first quarter of 2021. The response to the shocks reports volatility, particularly negative trends, until the fourth period, and despite a respite in the fifth period, further declines in periods six, seven and nine.

4.2.5. Construction

The construction sector includes establishments primarily engaged in constructing, repairing, renovating buildings, engineering, subdividing and developing land.[30] The sector accounted for 6% of employment in Toronto in 2020, the proportion that remained stable despite the pandemic contractions (Table 3). Significant business closures occurred in 2020, especially during April, May and June, during which business closings as a percentage of active businesses jumped to 18%, 13% and 9%, respectively (Table A1, Appendix A).

This sector employed approximately 97,430 individuals in 2020. However, a sharp decline followed the COVID-19 closures, with job losses concentrated in April (−10%) and May (−14%) (Figure 18). The losses were much more profound in the specialty trade subsector that bore the brunt of the crisis due to households delaying construction projects out of the fear of virus transmission and social distancing rules (Figure 19). The year ended with a net loss of 81,000 jobs in this sector or a decline of 8% from 2019 (Table 2). The IRF shows a lag in the pandemic's shock, with five negative quarters (Figure 20 and Table A2, Appendix A).

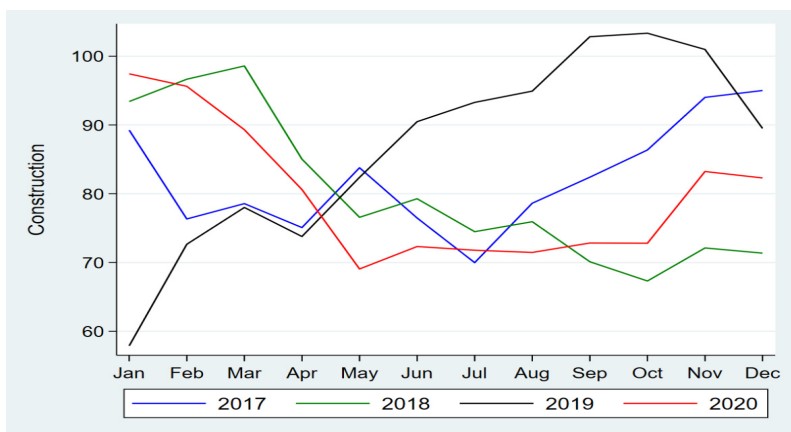

**Figure 18.** Employment in Construction sector. The employment in this sector stabilised towards the end of 2020, despite the decline in the initial period of the start of the pandemic. The y-axis of the figures represents the number of employees in 1000s. Source: Labour Force Survey, Statistics Canada.

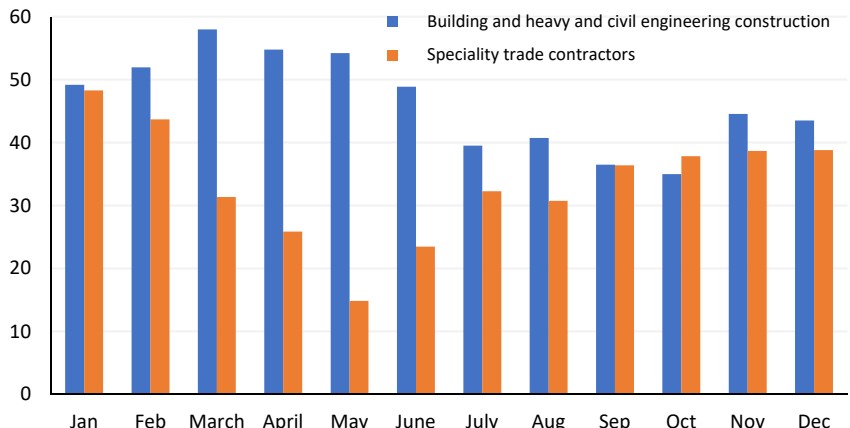

**Figure 19.** Construction—employment in key segments, 2020. The y-axis of the figures represents the number of employees in 1000s. Source: Labour Force Survey, Statistics Canada.

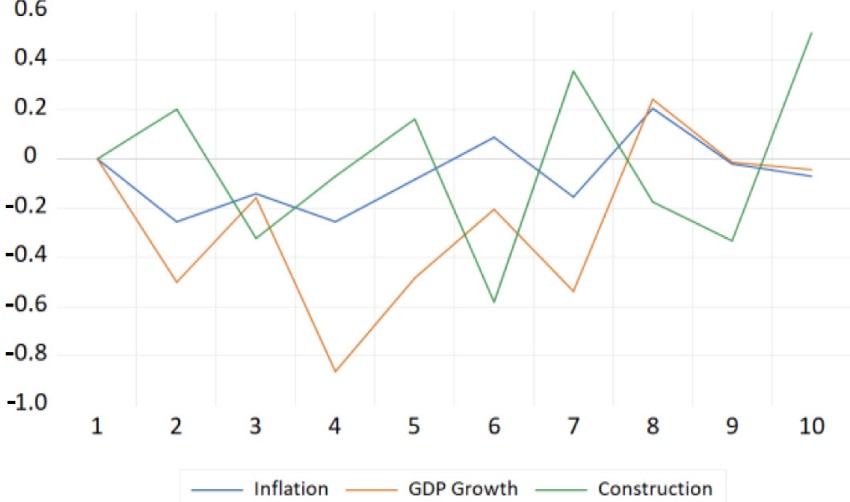

**Figure 20.** Construction—IRF[31]. The figure depicts the impact of inflation, GDP growth for Ontario, and GDP in the construction sector on unemployment in the sector, starting the first quarter of 2021. A lag in the pandemic's shock occurs with declines in employment in the third period and volatility in the proceeding periods.

*4.3. Stable Sectors*

4.3.1. Professional, Scientific and Technical Services

This sector comprises establishments primarily engaged in activities in which human capital is the major input.[32] It was the largest sector in Toronto in terms of employment, with a share of 14% of total jobs in 2020 (Table 3). It employed 200,012 workers at the start of the year, which fell to 69,056 in June and then climbed to 221,073, ending the year with a net annual increase of 1% (Table 2). The yearly employment exceeded the previous three years, substantiating this sector's strength (Figure 21). A likely reason for strength was that this sector comprises higher-paying legal, accounting, engineering, and other technical jobs. The ability to pivot towards virtual work was also a critical factor in the workforce's stability—approximately 45% of this sector could transition to virtual work, the highest among all sectors due to the pandemic (Figure A1, Appendix A). Even though the IRF estimate shows an increase in unemployment in the initial five quarters, convergence to a transitory trend occurs in the seventh quarter (Figure 22 and Table A2, Appendix A).

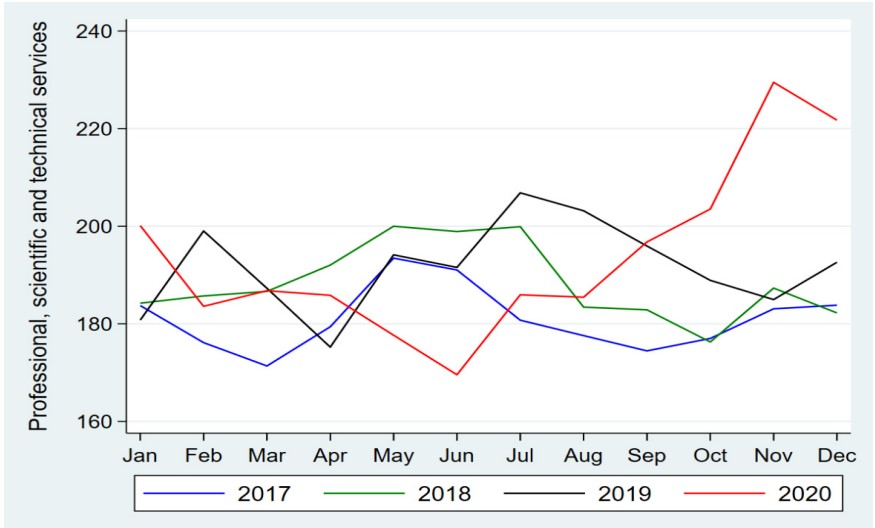

**Figure 21.** Employment in Professional, Scientific and Technical sector. The pandemic did not impact employment in this sector because of the transition to virtual work. The y-axis of the figures represents the number of employees in 1000s. Source: Labour Force Survey, Statistics Canada.

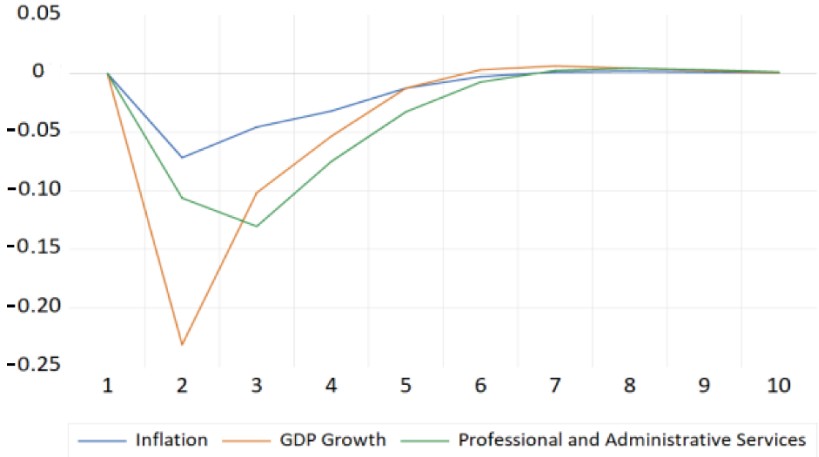

**Figure 22.** Professional, Scientific and Technical sector—IRF[33]. The figure depicts the impact of inflation, GDP growth for Ontario, and GDP in the professional, scientific & technical sector, starting the first quarter of 2021. Even though the IRF shows a drop in employment to the pandemic shock, convergence to a transitory trend occurs in the seventh period.

### 4.3.2. Finance, Insurance, Real Estate, Rental and Leasing (FIRE)

The FIRE sector comprises entities engaged in financial transactions—those involving the creation, liquidation, or change in ownership of financial assets—or facilitating financial transactions.[34] Historically, it has been one of Ontario's strongest sectors, accounting for two-thirds of its employees in the Toronto Economic Area (ER).[35] In 2020, it was the second-largest sector, accounting for 12% of all employment in Toronto (Table 3). This sector's profits are closely correlated to interest rates, household and business credit use, return on investments and economic stability. The year 2020 started with an employee count of 169,095 and ended at 178,052, increasing by 2%, the highest gain reported compared to other sectors (Figure 23 and Table 2). The annual employment outpaced previous years, except for 2019, with both major subsectors—finance & insurance and real estate & leasing—showing strong reliance in the face of the COVID-19 lockdowns (Figure 24). Many factors contributed to stable employment in this sector. The banking system's digitisation, the lack of equity stakes and lower claims in the insurance industry, and the continued sales growth in the detached housing market compensated for the sluggish numbers in the rentals and commercial real estate subsector. Approximately 25% of the finance and insurance workforce and 33% of the real estate subsectors could transfer to remote work due to the pandemic, one of the highest proportions among the sectors analysed (Figure A1, Appendix A).

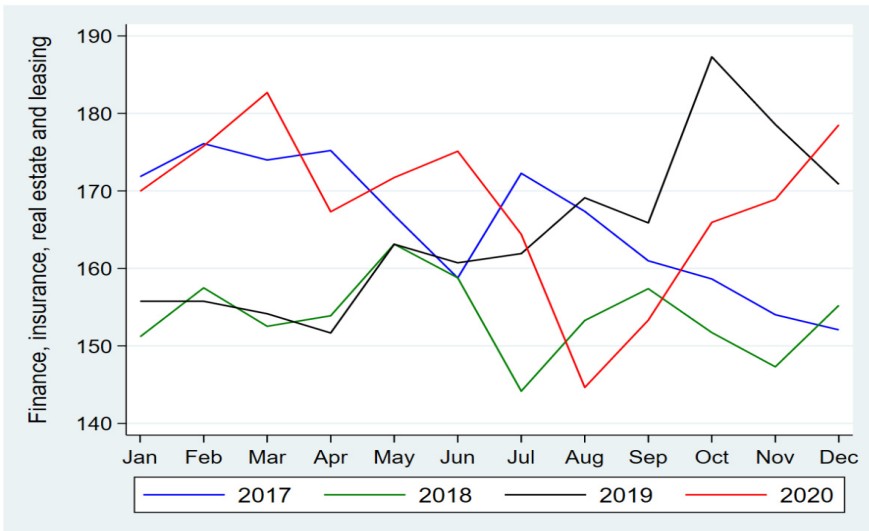

**Figure 23.** Employment in Finance, Insurance, Real Estate and Leasing sector. Despite a short-term drop in employment at the start of the pandemic in March, the levels recovered strongly, starting August. The y-axis of the figures represents the number of employees in 1000s. Source: Labour Force Survey, Statistics Canada.

In general, the insurance industry experienced little disruption during the pandemic. More than half of the industry was immune to declines in capital markets, as it held little or no equity in its portfolios, regardless of size (Grzadkowska 2020). The Office of the Superintendent of Financial Institutions (OSFI) also took measures to allow insurers to maintain work during the pandemic by postponing certain initiatives to allow insurers to focus on COVID-19.[36] Furthermore, a fall in the number of auto accidents during the lockdown led to fewer payouts by the auto insurance industry. As with the insurance industry, Toronto's financial and banking industry was relatively unaffected due to support from the Bank of Canada and technological advancements in the banking system.[37] The digitisation of online banking and the increased use of direct deposits, e-transfers, and other app-based banking enabled many employees in the financial subsector to work from home and simplified banking for consumers stuck at home (Keefe and Monas 2020).

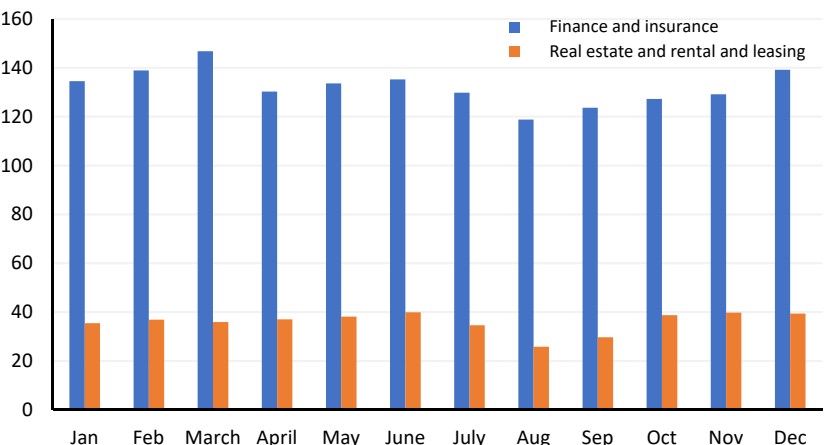

**Figure 24.** Employment in Finance, Insurance, Real Estate and Leasing sector—employment in key sub-sectors. The y-axis of the figures represents the number of employees in 1000s. Source: Labour Force Survey, Statistics Canada.

The real estate rental and leasing market experienced some fluctuations during the pandemic. The sales held steady during the pandemic but were not as strong as the Greater Toronto Area (GTA), which experienced a 24.3% increase from the previous year (Nanowski 2020). One residual effect of downtown residents' moving to the suburbs was the decline in the condo market. In November 2020, condo market sales increased marginally by 0.8%, while prices dropped by 3%, compared to an increase of 13% in sales in other residential housing market segments (Zivitz 2020). As with the impact of increased suburban home sales on the condo market, an increase in available rental units was visible as renters took advantage of lower interest rates by moving to the suburbs (Macdonell 2020). The increase in available rental homes led to a dramatic drop in rental prices, with the price of a one-bedroom apartment in Toronto falling by 20.4% in 2020. The commercial real estate market did not fare well during the pandemic. With telecommuting and virtual work at an all-time high, there was an increase in Canada's commercial vacancies of 10.8% in 2020 (Kerr 2020). The pandemic's impulse response shows that the negative trend in response disappears in the seventh quarter (Figure 25 and Table A2, Appendix A).

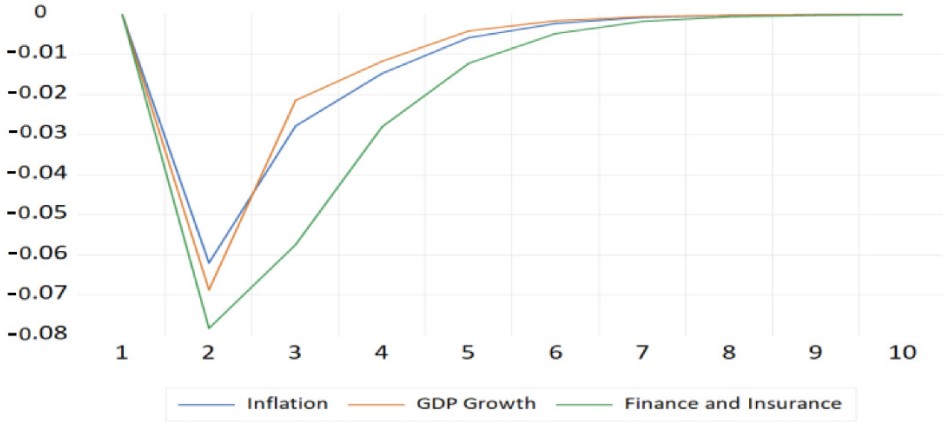

**Figure 25.** Finance, Insurance, Real Estate and Leasing sector—IRF[38]. The figure depicts the impact of inflation, GDP growth for Ontario, and GDP in the FIRE sector, starting the first quarter of 2021. The negative effect on employment disappears in the seventh quarter with the convergence.

## 5. Conclusions

The results of this study show that COVID-19 has had an uneven impact on the various sectors of Toronto's economy, with low-paying sectors suffering disproportionately,

leading to job losses. In particular, sectors such as accommodation & food services and manufacturing have suffered the most, with several others, such as retail, also facing a tremendous squeeze on revenues and viability. The results confirm similar findings in other countries such as Slovakia (Svabova et al. 2020), Portugal (Almeida and Santos 2020), Ireland (Byrne et al. 2020), the US (Bartik et al. 2020), UK (Harris et al. 2020) and Italy (Rapaccini et al. 2020).

While we limit our analysis to Toronto, we expect the impact to be similar in other areas of Canada, as evident by recent studies such as Roy et al. (2021), who analyse the effects of the pandemic on sectors such as Vancouver, another major city in Canada. The far-reaching impact of the pandemic means that targeted policy measures to help the ailing sectors are required. First, accommodation & food services and the manufacturing sector need assistance to deal with ongoing public health measures and a rapid increase in the use of technology at the expense of human labour. Second, adopting technology at the city level can increase the speed of new permit approval and recall of historical permit records to facilitate growth in the construction sector. Third, commercial tax breaks can address rising vacancies due to increased virtual work. Fourth, to attract residents back to the core, measures are needed to address the decline in commercial land usage, such as outdoor space development (parks and recreation). Fifth, repurposing commercial land to residential space, particularly affordable housing, can help increase housing affordability and reduce homelessness. Sixth, programmes for the most vulnerable segment of the population should focus on retraining to prepare for technology-oriented jobs in the future. Retraining and job readiness programmes can address the redundancy in organisations created by the increased use of technology. The impacts of the COVID-19 pandemic on Toronto will undoubtedly be widespread and unpredictable, but the focus remains on a strong recovery from the crisis despite this uncertainty. Navigating through the uncertainty of recovery can be positively supplemented by how the pandemic has forced changes in the various sectors of its economy. In guiding the city, policymakers need to understand these changes, trends, disruptions within sectors, and the effects on socio-economic groups.

**Author Contributions:** Conceptualization, V.S.; methodology, H.S. and V.S.; software, H.S.; validation, H.S.; formal analysis, V.S., H.S. and J.T.; investigation, V.S.; resources, V.S.; data curation, V.S. and H.S.; writing—original draft preparation, V.S.; writing—V.S., J.T., H.S. and review and editing, V.S. and J.T.; visualization, H.S.; supervision, V.S.; project administration, V.S.; funding acquisition, V.S. All authors have read and agreed to the published version of the manuscript.

**Funding:** This research was funded by MITACS and City of Toronto, grant number 1-51-47695. And The APC was funded by Ted Rogers School of Management, Ryerson University.

**Institutional Review Board Statement:** Not applicable.

**Informed Consent Statement:** Not applicable.

**Data Availability Statement:** Publicly available datasets were analyzed in this study. This data can be found here: https://www150.statcan.gc.ca/n1/pub/11-625-x/11-625-x2010000-eng.htm (accessed on 1 April 2021).

**Conflicts of Interest:** The authors declare no conflict of interest.

## Appendix A

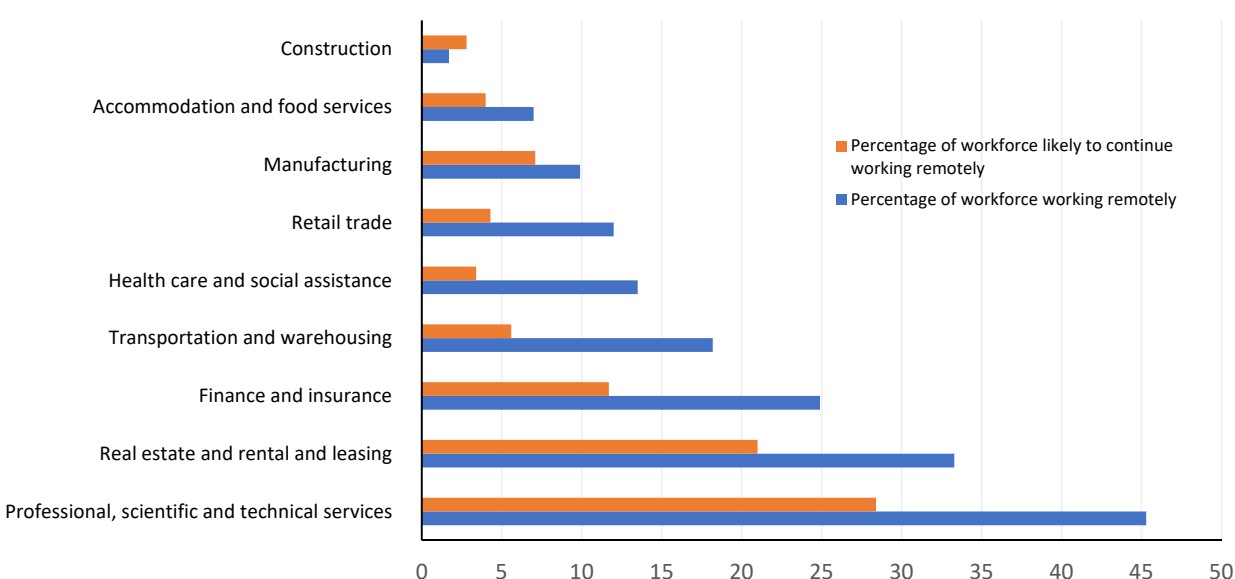

**Figure A1.** Remote Work (Toronto). Source: Canadian Survey of Business Conditions, Statistics Canada.

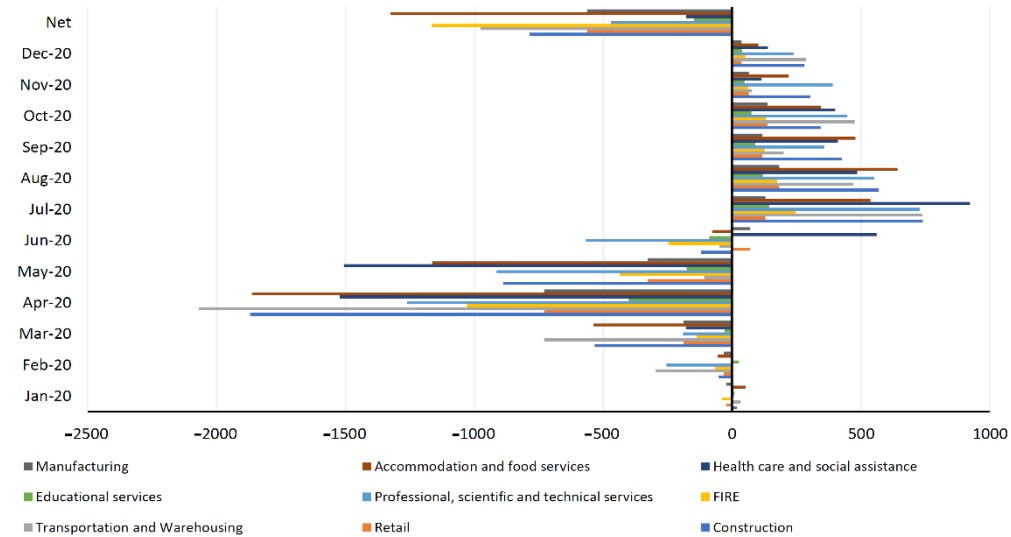

**Figure A2.** Bankruptcy (opening–closing businesses). Notes: (1) Opening businesses transition from having no employees in the previous month to having at least one employee in the current month. These instances occur when a new small firm begins, when a large firm opens a new establishment or when a seasonal firm reopens. (2) Closing businesses are those that transition from having at least one employee in the previous month to having no employees in the current month. These instances occur when a small firm goes out of business, when a large firm closes an establishment temporarily or permanently, and when a seasonal firm ceases business activity for the year. (3) Net: refers to the aggregate number for the year. Source: Longitudinal Employment Analysis Program, Statistics Canada.

**Table A1.** Closing businesses as a % of active business.

| | January-2020 | February-2020 | March-2020 | April-2020 | May-2020 | June-2020 | July-2020 | August-2020 | September-2020 | October-2020 | November-2020 | December-2020 | Avg. 2019 | Avg. 2019 | Avg. 2020 |
|---|---|---|---|---|---|---|---|---|---|---|---|---|---|---|---|
| Construction | 5% | 5% | 8% | 18% | 13% | 9% | 4% | 4% | 4% | 5% | 5% | 5% | 5% | 5% | 7% |
| Manufacturing | 3% | 3% | 4% | 12% | 8% | 5% | 4% | 2% | 2% | 2% | 3% | 3% | 3% | 3% | 4% |
| Retail | 4% | 3% | 6% | 22% | 15% | 10% | 3% | 3% | 3% | 3% | 4% | 4% | 4% | 4% | 7% |
| Transportation & warehousing | 8% | 8% | 11% | 24% | 10% | 10% | 7% | 7% | 7% | 7% | 7% | 7% | 8% | 8% | 9% |
| Finance and insurance and management of companies and enterprises | 5% | 5% | 5% | 11% | 7% | 6% | 5% | 5% | 5% | 5% | 5% | 5% | 5% | 5% | 6% |
| Real estate and rental and leasing | 5% | 5% | 6% | 14% | 11% | 10% | 4% | 5% | 5% | 6% | 6% | 6% | 5% | 5% | 7% |
| Professional, scientific and technical services | 6% | 6% | 6% | 11% | 9% | 8% | 5% | 5% | 5% | 5% | 5% | 6% | 6% | 6% | 7% |
| Educational services | 4% | 3% | 5% | 25% | 15% | 11% | 4% | 3% | 3% | 3% | 3% | 4% | 5% | 4% | 7% |
| Health care and social assistance | 3% | 3% | 4% | 13% | 14% | 7% | 3% | 2% | 2% | 3% | 3% | 3% | 3% | 3% | 5% |
| Accommodation & food services | 3% | 4% | 9% | 27% | 22% | 13% | 5% | 2% | 3% | 3% | 4% | 5% | 4% | 4% | 8% |

Notes: (1) The data is for Toronto. (2) Active businesses are those businesses that reported having one or more employees in a given month. Closing businesses are those that transition from having at least one employee in the previous month to having no employees in the current month. These instances occur when a small firm goes out of business, when a large firm closes an establishment temporarily or permanently, and when a seasonal firm ceases business activity for the year. Source: Longitudinal Employment Analysis Program, Statistics Canada.

**Table A2.** Impulse Response to COVID-19.

| Period | Accommodation & Food Services | Manufacturing | Retail Trade | Transportation & Warehousing | Educational Services | Construction | FIRE | Professional Scientific & Technical |
|---|---|---|---|---|---|---|---|---|
| 1 | 0 | 0 | 0 | 0 | 0 | 0 | 0 | 0 |
| 2 | −0.212 | −0.1068 | −0.0808 | −0.0792 | −0.0148 | 0.2006 | −0.0782 | −0.1062 |
| 3 | −0.0439 | −0.0689 | 0.0124 | 0.0863 | −0.0419 | −0.3233 | −0.0574 | −0.1303 |
| 4 | 0.1257 | −0.0554 | −0.1407 | 0.1026 | −0.0022 | −0.0718 | −0.028 | −0.0748 |
| 5 | 0.0937 | −0.0413 | −0.1163 | 0.0205 | 0.1889 | 0.1625 | −0.0122 | −0.0327 |
| 6 | −0.1261 | −0.0312 | −0.0532 | −0.0285 | −0.1248 | −0.5813 | −0.0048 | −0.0072 |
| 7 | −0.2412 | −0.0235 | −0.0456 | −0.0199 | −0.0178 | 0.357 | −0.0018 | 0.0029 |
| 8 | −0.1028 | −0.0178 | −0.0188 | 0.0025 | 0.0789 | −0.1734 | −0.0006 | 0.0047 |
| 9 | 0.1039 | −0.0134 | −0.0149 | 0.0089 | −0.0678 | −0.334 | −0.0002 | 0.0034 |
| 10 | 0.1133 | −0.0101 | −0.0174 | 0.0029 | 0.0161 | 0.5125 | −0.0001 | 0.0017 |

Note: The response refers to the impact on employment due to the COVID-19 shock starting in the first quarter of 2021. The periods refer to the quarters.

## Notes

1   Source: https://www.forbes.com/sites/bernardmarr/2018/09/02/what-is-industry-4-0-heres-a-super-easy-explanation-for-anyone/?sh=5b5d8a599788 (accessed on 15 September 2021).

2   Source: https://www.weforum.org/agenda/2021/11/how-tech-4-0-helped-companies-survive-covid-19/ (accessed on 11 December 2021).

3   The forcast from the VAR model for each sector starts from the fourth quarter of 2020 and lasts for 10 periods. The use of 10 period allows us to measure a reasonable future response to the shock. It should be noted that the use of such time horizon corresponds to standard practice in the literature, for example, see Darolles and Gourieroux (2015); Oscar Jordà (2005) and Koop et al. (1996).

4   We chose optimal lag of 1 according to AIC and SC information criteria for Accommodation & food services, Manufacturing, Retail trade, Transportation & warehousing, Educational services, Professional scientific & technical, and Finance, insurance, real estate and leasing sectors. For the Construction sector, the optimal lag is 2.

5   The same VAR model was estimated for each sector to obtain the Impulse Response Functions (IRFs).

6   We use monthly data from 2017 to 2020.

7   A cutoff criterion of 5% of total employment is used to identify the top sectors.

8   Source: https://www23.statcan.gc.ca/imdb/p3VD.pl?Function=getVD&TVD=118464&CVD=118465&CPV=72&CST=01012012&CLV=1&MLV=5 (accessed on 15 March 2021).

9   Source: https://www.jobbank.gc.ca/content_pieces-eng.do?cid=12204 (accessed on 15 March 2021).

10  Ibid.

11  The responses of unemployment to Cholesky One S.D (d. f. adjusted) Innovations $\pm$ 2 S. E.

12  Source: https://www23.statcan.gc.ca/imdb/p3VD.pl?Function=getVD&TVD=118464&CVD=118465&CPV=31-33&CST=01012012&CLV=1&MLV=5 (accessed on 15 March 2021).

13  Source: https://www150.statcan.gc.ca/n1/daily-quotidien/200715/dq200715a-eng.htm (accessed on 17 March 2021).

14  The responses of unemployment to Cholesky One S.D (d. f. adjusted) Innovations $\pm$ 2 S. E.

15  Source: https://www23.statcan.gc.ca/imdb/p3VD.pl?Function=getVD&TVD=118464&CVD=118465&CPV=44-45&CST=01012012&CLV=1&MLV=5 (accessed on 15 March 2021).

16  Retail e-commerce sales in Canada increased from 2.9 billion in July 2020 to 4.3 billion in November 2020. Source: https://www150.statcan.gc.ca/t1/tbl1/en/tv.action?pid=2010007201 (accessed on 15 March 2021).

17  Source: https://www15.0.statcan.gc.ca/t1/tbl1/en/tv.action?pid=2010007401 (accessed on 17 March 2021).

18  The responses of unemployment to Cholesky One S.D (d. f. adjusted) Innovations $\pm$ 2 S. E.

19  Source: https://www23.statcan.gc.ca/imdb/pD.pl?Function=getVD&TVD=118464&CVD=118465&CPV=48-49&CST=01012012&CLV=1&MLV=5 (accessed on 15 March 2021).

20  Source: https://www.jobbank.gc.ca/content_pieces-eng.do?cid=14662 (accessed on 15 March 2021).

21  Source: https://www.cbc.ca/radio/asithappens/as-it-happens-Tuesday-edition-1.5851623/ontario-lockdown-doesn-t-do-anything-to-help-warehouse-workers-says-advocate-1.5852114 (accessed on 15 March 2021).

22  Source: http://www.ttc.ca/Coupler/Editorial/Notices/index.jsp (accessed on 20 March 2021).

23  Source: https://travel.gc.ca/travel-covid/travel-restrictions/border (accessed on 20 March 2021).

24  The responses of unemployment to Cholesky One S.D (d. f. adjusted) Innovations $\pm$ 2 S. E.

25  Source: https://www23.statcan.gc.ca/imdb/p3VD.pl?Function=getVD&TVD=118464&CVD=118465&CPV=62&CST=01012012&CLV=1&MLV=5 (accessed on 15 March 2021).

26  Source: https://www23.statcan.gc.ca/imdb/p3VD.pl?Function=getVD&TVD=118464&CVD=118465&CPV=61&CST=01012012&CLV=1&MLV=5 (accessed on 15 March 2021).

27  Source: https://www.jobbank.gc.ca/content_pieces-eng.do?cid=15197 (accessed on 15 March 2021).

28  Source: https://www150.statcan.gc.ca/n1/daily-quotidien/201009/dq201009a-eng.htm (accessed on 17 March 2021).

29  The responses of unemployment to Cholesky One S.D (d. f. adjusted) Innovations $\pm$ 2 S. E.

30  Source: https://www23.statcan.gc.ca/imdb/pD.pl?Function=getVD&TVD=118464&CVD=118465&CPV=23&CST=01012012&CLV=1&MLV=5 (accessed on 15 March 2021).

31  The responses of unemployment to Cholesky One S.D (d. f. adjusted) Innovations $\pm$ 2 S. E.

32  Source: https://www23.statcan.gc.ca/imdb/p3VD.pl?Function=getVD&TVD=118464&CVD=118465&CPV=54&CST=01012012&CLV=1&MLV=5 (accessed on 15 March 2021).

33  The responses of unemployment to Cholesky One S.D (d. f. adjusted) Innovations $\pm$ 2 S. E.

34  Source: https://www23.statcan.gc.ca/imdb/p3VD.pl?Function=getVD&TVD=118464&CVD=118465&CPV=52&CST=01012012&CLV=1&MLV=5 (accessed on 15 March 2021).

35    Source: https://www.jobbank.gc.ca/content_pieces-eng.do?cid=14525 (accessed on 15 March 2021.

36    Source: https://www.osfi-bsif.gc.ca/Eng/Pages/COVID-19.aspx (accessed on 19 March 2021).

37    Source: https://www.bankofcanada.ca/2020/05/financial-system-review-2020/ (accessed on 17 March 2021).

38    The responses of unemployment to Cholesky One S.D (d. f. adjusted) Innovations $\pm$ 2 S. E.

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
