# Peer review of "The COVID-19 Era—Influencers of Uneven Sector Performance: A Canadian Perspective"

_economies, doi:10.3390/economies10020040_

Round 1

Reviewer 1 Report

The focus of the paper is current. The processing procedure and the applied methods are, in my opinion, appropriately chosen. However, certain parts of the paper may be improved, they show inconsistencies or the graphical processing of the results is not clear and may be improved. I suggest you to:

1.) to incorporate in the introduction section more research papers presenting the impact of COVID-19 on the labor markets (Svabova, L., & Gabrikova, B. (2021). The rise in youth employment? Impact evaluation of COVID-19 consequences. Journal of Eastern European and Central Asian Research (JEECAR)8(4), 511-526. https://doi.org/10.15549/jeecar.v8i4.757 ; Svabova, L., Metzker, Z., Tomasz, P. (2020). Development of unemployment in Slovakia in the context of the COVID-19 pandemic, Ekonomicko-manazerske spektrum, 14(2), 114-123. https://doi.org/10.26552/ems.2020.2.114-123; Svabova, L., Kramarova, K. (2021). An analysis of participation factors and effects of the active labour market measure Graduate practice in Slovakia – Counterfactual approach. Evaluation and Program Planning. 86. https://doi.org/101917. 10.1016/j.evalprogplan.2021.101917; and/or Svabova, L., Kramarova, K., & Durica, M. (2021). Evaluation of the Effects of the Graduate Practice in Slovakia: Comparison of Results of Counterfactual Methods. Central European Business Review10(4), 1-31. https://doi.org/10.18267/j.cebr.266 etc.)

2.)  in the part Methodology you present the equations for 4 IRFs, however in each picture where the IRFs are depicted, you present only 3 of them - why?  

3.) in the same part you present the IRF for unemployment (the first equation for IRFS), however in the text and in the figures you talk about IRF for employment.

4.) in the figures where are IRFs depicted, you use designation of the analysed sector (e.g. Fig. 3 green line presents "Accomodation and food services"), however is not exactly known if you depict IRF for unemployment or IRF for the growth rate of the sector. TThe same is true for other sectors and the IRF curves shown.

5.) in the figures where the employment for each sector is depicted (line charts), the used unit of measure is mentioned only in the text - I suggest you to put it also directly in the figures.  

6.) in the figures where the employment for each sector is depicted (bar graphs) on a monthly basis, I am missing the time period (year). There is also the description of the y-axis missing.

7.) I suggest you to move from the Appendix part the information on the structure of employment based on sectors (Table A1) into the part "Results".  

Reviewer 2 Report

The VAR model (lines 98 to 104) should be written in a more rigorous way; all the parameters a, b, c, d and f, are represented using the same letter (without an equation subscript); also the random errors for each equation should be different. The three quarterly dummy variables are dependent also on t, subscript which is missing. The number of lags, k, is not specified. So from this formula, where the central model is used, the reader can not have a clear idea of what are the real aims of the paper.

At the beginning of the methodology, data should be commented; the reader has to deduce that they are quarterly data from the seasonal dummy variables. But later on, in figure 1 and most of the following figures, the data used are monthly. So, it is necessary to describe briefly all data used from the start. It seems from the different figures presented that available data are monthly for 4 years, up to 2020, and quarterly forecasts are from 2021 for 10 quarters, that is 2.5 years. Or, in one looks at table A3 about the VAR model, there are 10 periods of quarterly lags. If this is so, there are data of 16 quarters (2017-20), with 10 lags which leaves 6

It is not clear (lines 107-108) the sentence: 'Sectort represents the growth rate of the industry I'. What is I?      

In lines 186 to 188 some trivial matters are explained; it is evident that the x-axis represents the months, while the y-axis explanation is not clear: these are the number of employees (measures in 103)? What is the 'number of employees per 100000'.

In figure 2, there is no reference about the unit used (thousands again?).

The same for figure 3; what are the meaning? Proportions? Was there a fall of over 40% in GDP? Or are the time series normalized? In this case the x-axis seems to represent quarters; if this is so, it would be more informative to specify the years and quarters.

Also, this figure is labelled 'Accommodation and food services', which surely means employment in this sector (well, the rate of changes, which is not specified).

But figure 3 is representing also some additional variables; if the first 3 quarters are 2021 data, the rest are forecasts obtained with the VAR model. Or maybe all are forecasts? It is not clear what data are used and what forecasts are obtained.

The same can be commented about figures 6, 9, 11, 17, 20, 22 and 25.

Reviewer 3 Report

Dear Authors

I am very happy to have the opportunity to read your interesting article. I hope that a few of my remarks will help to make it even better. Here they are:

  1. The manuscript entitled “The COVID-19 era: Influencers of uneven sector performance on labour market outcomes“ is an analytical case study written at the correct scientific level. The authors should be appreciated for the choice of topics for their deliberations and the attempt to objectively look at the impact of COVID-19 on the labour market outcomes of major industrial sectors in Toronto, the largest urban centre in Canada.
  2. The title of the work should be modified. Why "The COVID-19 era"? The pandemic lasts just over two years. The authors analyse the data mainly for 2020 and are already writing about the era? It is too much of an exaggeration. Please be more careful with words.
  3. In my opinion, the work should have clearly separated, independent sections: Introduction and Literature review. In the Introduction section, please present the aims of the article, research hypotheses or research questions. Moreover, please write to whom the article is directed. Finally, please add a separate paragraph to describe the structure of the article.
  4. The literature review section is essential. The authors should present the current state of knowledge on the topic they are discussing.
  5. The Discussion section should be necessarily enriched with conclusions from research conducted by other researchers. The material presented in the results section is very rich. Now, in the Discussion section, this material should be confronted with the results of similar studies described in the literature on the subject. After this confrontation of the authors' results with those of other researchers, the conclusions should be drawn.
  6. Researchers do not recognize the limitations of their research. Can the results obtained for Toronto apply to the economy of Canada as a whole? Can a similar situation be expected in other parts of the world?
  7. What theoretical and practical implications does the text prepared by the authors have? It would be worth mentioning them in the Conclusion section, which should be separated from the Discussion.
  8. References are very relevant and up-to-date, but very poor at the same time. They need to be expanded.

I hope that the indicated remarks will help the Authors to improve their text so that the work will be published. Good luck!

Round 2

Reviewer 2 Report

page 3 note 3: correct 'fourth'

page 3: subscripts of the VAR model: some are written as (t - i) and some others as t - i; Also the variable GDPSector seems to be written either as GNPSector or Sector;

page 3 on line 124, it should be sector i (lowercase)

page 3: How many lags are in the VAR model? 10?

In the methodology, quarterly data are from the first quarter of 2000 to the third of 2020; please state this clearly in line 108.

In note 3, the forecast period is specified at 10 quarters, which is somewhat optimist

page 21: in the impulse-response function, why there is no coefficient estimated with lag 1.

Reviewer 3 Report

Dear Authors

Despite the fact, that the fulfillment of my recommendations did not meet all of my expectations, in the current version the article can be published.

Best regards

Reviewer

Author Response

We thank the Reviewer.